# Identification of Molecular Subtypes and Prognostic Traits Based on Chromosomal Instability Phenotype-Related Genes in Lung Adenocarcinoma

**DOI:** 10.3390/cancers16223818

**Published:** 2024-11-13

**Authors:** Youngsic Jeon, Kyung-Chul Choi, Young Nyun Park, Young-Joo Kim

**Affiliations:** 1Institute of Natural Products, Korea Institute of Science and Technology, Gangneung 25451, Republic of Korea; biomangg0@kist.re.kr; 2Department of Biochemistry and Molecular Biology, Brain Korea 21 Project, Asan Medical Center, University of Ulsan College of Medicine, Seoul 05505, Republic of Korea; choikc75@amc.seoul.kr; 3Department of Pathology, Graduate School of Medical Science, Brain Korea 21 Project, Yonsei University College of Medicine, Seoul 03722, Republic of Korea; young0608@yuhs.ac

**Keywords:** lung adenocarcinoma, chromosomal instability, prognostic marker

## Abstract

This study provides evidence that LUAD-significant CIN-related genes, identified through transcriptomic profiling, serve as a predictor of LUAD prognosis. By classifying CIN-related subtypes based on their gene expression stratifications (Group^Low^, Group^Moderate^, and Group^High^), it links these subtypes to distinct survival outcomes, with the Group^High^ showing poor prognosis, as well as transcriptomic and genomic alterations. These findings, validated across multiple cohorts, highlight the significance of CIN-related gene status in guiding future clinical interventions.

## 1. Introduction

Lung adenocarcinoma (LUAD), the most common subtype of non-small cell lung cancer, represents a significant clinical challenge due to its molecular complexity and heterogeneity [1]. Despite advances in therapeutic strategies, LUAD remains associated with high morbidity and mortality rates globally, with survival outcomes often linked to the genetic and molecular landscapes [2,3]. Many studies to classify LUAD have largely focused on identifying driver mutations such as *EGFR*, *ALK*, *KEAP1*, *TP53*, and *KRAS*, leading to the development of targeted therapies [2,4,5]. However, these approaches have not fully accounted for the extensive molecular diversity within LUAD, particularly regarding chromosomal instability (CIN) and its implications for prognosis.

Chromosomal instability (CIN), a hallmark of cancer, refers to the high rate of chromosomal gains and losses during cell cycle and division, leading to aneuploidy and genomic alterations [6,7]. This phenomenon is driven by various factors, including chemical and radical exposure, smoking, viral infections, oxidative stress, and inflammation [8,9]. Many studies have suggested that CIN plays a critical role in tumor progression, metastasis, and therapy resistance in various cancers, including LUAD and hepatocellular carcinoma (HCC) [10]. While many studies have focused on gene signatures to define molecular subtypes associated with patient survival in various cancers, similar research in LUAD remains relatively limited. Moreover, the identification of specific CIN-related genes and their potential as prognostic markers in LUAD has not been fully explored.

In this study, we performed transcriptomic profiling using TCGA-LUAD data and identified twenty-four LUAD-significant CIN-related genes associated with tumor grade in LUAD patients to explore potential biomarkers for disease progression. Among these, we found a subset of genes related to the CIN phenotype, suggesting that CIN phenotype gene expression may be associated with tumor aggressiveness and grade in LUAD. From this, we first reviewed the transcriptomic and genomic traits in LUAD. Our findings not only highlight the prognostic significance of CIN-related genes status in LUAD but also provide a framework for future clinical validation and potential therapeutic strategies aimed at improving patient outcomes.

## 2. Materials and Methods

### 2.1. Transcriptome Data Analysis

Public data from The Cancer Genome Atlas-Lung adenocarcinoma (TCGA-LUAD, *n* = 594) dataset, consisting of normal tissues (*n* = 59), tumors (*n* = 533), and recurrent tumors (*n* = 2), were obtained using the TCGAbiolinks package in R (version, 4.2.2). Recurrent tumors were excluded from the analysis. For validation, the GSE42127 (*n* = 176), GSE37745 (*n* = 196), GSE31210 (*n* = 246), and GSE50081 (*n* = 181) datasets were retrieved from the GEO database. For data integration, the transcriptome levels of each cohort were calculated using log_2_ transformation, and these transcriptome levels were merged by gene symbols. Subsequently, a pooled dataset was performed by correcting batch effects using the “Combat” library in R.

To calculate the enrichment scores of gene signatures, gene sets were obtained from the Gene Ontology (geneontology.org, accessed on 12 November 2022) and Reactome (www.reactome.org, accessed on 12 November 2022) databases. The classification of LUAD patients of a pooled dataset (*n* = 779), based on the established differentially expressed genes, was performed using the nearest template prediction (NTP) algorithm, with a false discovery rate (FDR) < 0.05 as the threshold for statistical significance, as described previously [11].

The oncoactivity score was determined by calculating the differential enrichment of oncogene activation and tumor suppressor gene (TSG) repression. The formula used was as follows: Oncoactivity = ES_oncogene_ − ES_TSG_, where ES refers to the enrichment score for each gene set. The lists of oncogenes (*n* = 674) and TSGs (*n* = 1088) were obtained from previous studies [12].

### 2.2. Mutation Profiling

For mutation profiling, the TCGA-LUAD dataset (*n* = 509) was used, focusing on somatic variants in exon regions. Non-significant variants were filtered out based on the following criteria. Allele frequencies of the variants were obtained from the normal populations, including the 1000 Genome Project and Exome Aggregation Consortium dataset. Non-significant variants were filtered out before analysis, including variants with mutated read counts less than 8, mutation frequencies greater than 30%, or missing values in more than 30% of the samples. Additionally, mutations with an allele frequency greater than 5% in either of the normal populations were excluded. This filtering process resulted in the identification of 10,556 somatic exon mutations.

### 2.3. Analysis of Copy Number Variation

For copy number variation (CNV) profiling, the TCGA-LUAD dataset (*n* = 514) was used. Gene-level DNA copy numbers of each sample were mapped to their corresponding segment aberration values. Probes with more than 20% of missing values across samples were filtered out, and the remaining missing values were imputed. Copy number gains and losses were identified using thresholds of 0.5 and −0.5, respectively.

### 2.4. Statistical Analysis

Statistical analysis was conducted using R software (version 3.4.0; Vienna, Austria). Statistical significance was assessed using one-way ANOVA and *Chi*-squared test. All *p*-values of less than 0.05 were considered statistically significant. All figures were created using RStudio (version 2022.7.2.576; RStudio, Inc., Boston, MA, USA).

## 3. Results

### 3.1. Chromosomal Instability Phenotype-Related Genes Are Associated with Pathologic Stage in Lung Adenocarcinoma

Transcriptome profiles of LUAD, including 59 cases of normal tissue and 533 cases of LUAD, were obtained from the TCGA-LUAD (*n* = 592), a well-organized cohort with comprehensive clinical data. To investigate the underlying mechanisms associated with the aggressive traits of LUAD depending on its pathologic grade [American Joint Committee on Cancer (AJCC)], we performed transcriptomic profiling using the Gene Ontology (GO:BP, *n* = 2120) database. Notably, OS was significantly associated with pathological grade compared to other clinical features (*p*-value < 10^−5^, Figure 1A, top). By performing unsupervised clustering of gene signatures, we found that cell cycle- (10/27, 37.04%) and DNA replication-related signatures (6/27, 22.22%) were predominantly enriched in a stage-dependent manner (*p*-value < 0.01, one-way ANOVA, Figure 1A, middle, and Appendix A). Among these signature-related genes, we focused on 24 genes related to chromosomal instability (CIN), selected based on the CIN70 signature proposed, which identified markers predictive of clinical outcomes across multiple human cancers [13]. These 24 genes were significantly increased in relation to pathologic grade (*p*-value < 0.05, one-way ANOVA; fold change (FC) > 0.3 compared to IA grade, Appendix A) and were found to be associated with oncogenes (e.g., *CCNB1*, *PLK1*, *CDC6*, *CDK1*, etc., *n* = 12). Using these 24 genes, we analyzed patient prognosis across 31 cancer types in the TCGA dataset and found associations with patient outcomes in 4 cancer types, including LUAD (*p*-value < 0.001, Appendix A). Based on these findings, we considered this an LUAD-significant signature. Congruently, the mean expression levels of LUAD-significant CIN phenotype genes were noticeably increased in high-grade compared to low-grade in other previous cohorts (*p*-value < 0.01, one-way ANOVA, Figure 1B).

### 3.2. LUAD-Significant CIN Phenotype-Related Gene Status Revealed an Association with Transcriptomic Patterns and Prognostic Predictions

To evaluate the molecular characteristics associated with LUAD-significant CIN phenotype-related gene expression, LUAD patients were divided into three groups, low (Group^Low^, *n* = 176), moderate (Group^Moderate^, *n* = 176), and high (Group^High^, *n* = 181), based on stratifications of LUAD-significant CIN phenotype-related gene expression levels, using the quantile value as the cutoffs (low ≤ 33%, 33% < moderate ≤ 66%, high > 66%, Figure 2A). By performing unsupervised clustering using variably expressed genes (median absolute deviation [MAD] > 0.6, *n* = 4941), we found that the transcriptomes of the LUAD patients were clearly classified based on the CIN phenotype-related gene status (*p*-value < 10^−15^, Figure 2B). Principal component analysis also revealed that all tumor samples were distinctly separated from normal tissues, reflecting the striking distribution between groups (Figure 2C). Additionally, we evaluated the association between groups defined by CIN phenotype gene status and LUAD patient survival. Correspondingly, Kaplan–Meier survival analysis demonstrated that the Group^High^ patients exhibited poor overall survival [OS, hazard ratio (HR) = 1.43, *p*-value < 10^−3^] and disease-free survival (DFS, HR = 1.27, *p*-value = 0.0057) (Figure 2D,E). The Group^High^ patients also exhibited a significant association with age (*p*-value < 0.01), gender (*p*-value < 0.001), and pathologic grade (*p*-value < 0.001) (Appendix A). Moreover, we conducted univariate and multivariate analyses using the TCGA-LUAD cohort and found that the expression of CIN-related gene status (HR = 2.18, *p*-value < 0.001) and pathologic grade (HR = 1.2, *p*-value < 0.001) were an independent prognostic predictor of the clinical outcomes in LUAD patients (Appendix A). These results suggest that LUAD-significant CIN phenotype-related gene status may be associated with potential markers for disease progression in LUAD.

### 3.3. Characterization of Transcriptomic Subtypes According to LUAD-Significant CIN Phenotype-Related Gene Status

We identified differentially expressed genes [DEGs, *n* = 1534, *p*-value < 0.01, and FC > 1] between groups (Figure 3A, top, and Appendix A), which were categorized into “DEG low” (*n* = 799), “DEG moderate” (*n* = 18), and “DEG high” (*n* = 717). Among these, the Group^High^ showed the significant enrichment of cell cycle-related signatures, whereas the Group^Low^ was enriched in metabolism-related signatures, such as surfactant pathways (Figure 3A, bottom). When we evaluated oncogene and tumor suppressor gene (TSG) profiling based on the DEGs, we observed a higher distribution of oncogenes but lower distribution of TSGs in the Group^High^ compared to the Group^Low^ (Figure 3B). Additionally, by calculating the oncoactivity score and counterbalanced suppression of the TSGs [i.e., enrichment score (ES)_oncogenes_—ES_TSG_] as described previously [12], we demonstrated a significantly higher oncoactivity score in the Group^High^ compared to the Group^Low^ (*p*-value < 0.001, Figure 3C). Next, we further analyzed the expression profiling of surfactant metabolism-related genes. Indeed, surfactant metabolism has been associated with predicted markers of better survival outcomes in patients with LUAD [14,15,16]. Notably, surfactant metabolism-related genes (e.g., *SFTPC*, *SFTPD*, *ABCA3*, *SFTPA2*, etc., *n* = 13) were significantly upregulated in the Group^Low^ compared to other groups (*p*-value < 0.001, Figure 3D), revealing a correlation between high expression levels and favorable prognosis in TCGA-LUAD and four independent cohorts (Appendix A). These findings suggest that LUAD-significant CIN phenotype-related gene status may be a strong indicator of poor prognostic outcomes in LUAD and is associated with distinct transcriptomic alterations.

### 3.4. Validation of the Molecular and Clinical Significance of LUAD-Significant CIN Phenotype-Related Gene Status in a Pooled LUAD Dataset

To obtain robust findings with an expanded sample size, we pooled independent cohorts from GEO (GSE42127, GSE37745, GSE50081, and GSE31210, *n* = 779) (Figure 4A). Using the nearest template prediction (NTP) algorithm based on previously established DEGs, we classified the pooled dataset into four groups; High-like [false discovery rate (FDR) < 0.05, *n* = 315], Moderate-like (*n* = 58), Low-like (*n* = 321), and a not determined (ND) group (*n* = 85, FDR ≥ 0.05, Figure 4B). We confirmed that LUAD-significant CIN phenotype-related genes were expressed at significantly higher levels in the High-like group compared to the other groups (*p*-value < 10^−15^, Figure 4C). Additionally, Kaplan–Meier survival analysis revealed the prognostic significance of CIN phenotype-related gene status in the independent cohorts (*n* = 4), showing an association between the High-like group and poor prognosis compared to the Low-like group (Figure 4D). Furthermore, surfactant metabolism-related genes were significantly upregulated in the Low-like group compared to the other groups across all independent cohorts (*p*-value < 10^−15^, Figure 4E). Considering these results collectively, we suggest that CIN phenotype-related gene status serves as a major predictor of poor survival outcomes, potentially linked to the depletion of surfactant metabolism. However, further studies are required to validate these findings in clinical studies.

### 3.5. Distinct Genomic Profiles According to LUAD-Significant CIN Phenotype-Related Gene Status

Next, we evaluated the mutation profiles of the CIN groups in the TCGA-LUAD cohort (*n* = 509). Overall, tumor mutation burdens differed across the subtypes, with more frequent mutations, including frameshift deletions, nonsynonymous mutations, stop gains, and stop losses, observed in the Group^High^ compared to the Group^Moderate^ and Group^Low^ (*p*-value < 0.05, Figure 5A), which might be linked to the differing CIN-related gene status across these groups. Noticeably, we demonstrated that mutations in *KEAP1* (82/509, 16.1%), *LYST* (44/509, 8.12%), *SETD2* (27/509, 5.3%), and *TP53* (17/509, 3.3%) were frequent in the Group^High^ but not in normal tissue or the Group^Low^ (*p*-value < 0.01, *Chi*-squared test, Figure 5B). In particular, we focused on that the *TP53* and *KEAP1* mutation frequency in the Group^High^ might be associated with cell cycle traits. Indeed, *KEAP1* and *TP53* mutations specifically focused on nonsynonymous regions, with a higher prevalence of these mutations in LUAD patients with poor prognosis [17,18]. Thus, we suggest that LUAD-significant CIN phenotype-related gene status is associated with *KEAP1* and *TP53* mutations, although further studies are needed to confirm these findings.

In the copy number variation (CNV) analysis, we observed that the Group^High^ exhibited a significant alteration in DNA copy numbers compared to normal tissues; notably, chromosome 3q, 5q, 8q, 12q, and 22q showed more profound and typical chromosomal gains and losses compared to Normal and the Group^Low^ (Figure 5C). To conduct a genome-wide evaluation across the groups, we assessed CNVs in the gene sets of oncogenes and TSGs. We observed that oncogenes preferentially exhibited copy number gains in the Group^High^, while no significant alterations were found for TSGs (Figure 5D).

### 3.6. Transcription Factor Profiles According to LUAD-Significant CIN Phenotype-Related Gene Status

We also conducted transcription factor (TF) profiling of LUAD-significant CIN-related genes, which were identified as a predictive signature in LUAD. To assess TF-wide alterations across the groups, we evaluated significant differences in the TF gene set (*n* = 1640) [19] and identified TFs (*n* = 44) with significantly higher expression and associated with poor outcomes (*p*-value < 0.01 and HR > 0.5). These TFs also showed elevated expression compared to normal samples (*p*-value < 0.01, Figure 6A). Among them, nine TFs were significantly upregulated in association with CIN phenotype-related gene status (*p*-value < 10^−5^, Figure 6B). Next, to delineate the key TFs modulating LUAD-significant CIN-related genes, we constructed a genetic network to determine their functional interactions. Our findings revealed that *FOXM1* (32/68) and *HMGA1* (13/68) potentially serve as central hubs, bridging CIN phenotype-related genes more prominently than other TFs (Figure 6C). Additionally, we found that expression of *FOXM1* and *HMGA1* was noticeably increased in a tumor grade-dependent manner compared to other TFs (*p*-value < 0.001, one-way ANOVA, Figure 6D and Appendix A). Indeed, immunohistochemistry analysis revealed that FOXM1 and HMGA1 protein levels were significantly elevated in high-grade patients compared to low-grade patients, with localization in the nucleus, reflecting their potential roles as TFs (Figure 6E,F).

## 4. Discussion

In this study, we conducted integrative analyses of the LUAD transcriptome and genome from the perspective of LUAD-significant CIN phenotype-related genes identified through transcriptomic profiling from the TCGA-LUAD dataset. We demonstrated that their expression status was significantly associated with prognosis in LUAD patients across independent cohorts, revealing distinct traits of both transcriptomic and genomic levels (summarized in Figure 7). Furthermore, our findings suggest that the expression of these genes may serve as a more precise predictive marker compared to other clinical traits such as age, gender, and pathologic grade. However, further studies are needed to validate these findings in clinical studies. Although many studies have investigated gene signatures to define molecular subtypes linked to prognostic prediction in various cancers such as HCC, comparable research in LUAD remains relatively limited. We believe this study has addressed that issue to some extent.

In this study, we performed transcriptomic profiling using TCGA-LUAD data and identified twenty-four LUAD-significant CIN phenotype-related genes. It is important to note that the 24 CIN phenotype-related genes identified in this study are based on the CIN70 signature described [13], which reflects segmental aneuploidy rather than a direct measurement of CIN itself. Additionally, we use the term ‘CIN phenotype’ as an umbrella term to encompass various cellular characteristics associated with chromosomal instability, including aneuploidy, tolerance to chromosomal aberrations, and increased cellular heterogeneity. Many of the LUAD-significant CIN phenotype-related genes (e.g., *CCNB1*, *CCNB2*, *CDK1*, *MYBL2*, and *MAD2L1*) are associated with the dimerization partner (DP), retinoblastoma (RB)-like, E2F and MuvB (DREAM) complex, which plays a critical role in the cell cycle, particularly in controlling the transition between quiescence (G0) and the active phases of the cell cycle (G1, S, G2, and M). Among them, *MYBL2* is a key molecule involved in the transition between the S and G2 phases of the cell cycle and is also a significant prognostic marker [20]. Malfunctions in the DREAM complex have been implicated in various cancers, including LUAD [21,22]. Consistently, we demonstrated that LUAD patients with poor prognosis showed the distinct enrichment of cell cycle- and DNA replication-related signatures.

Notably, the differential expression of surfactant metabolism-related genes in the Group^Low^, coupled with favorable survival outcomes, implies that CIN-related subtypes exhibit functional heterogeneity that may have therapeutic implications. Previous studies have shown that surfactant metabolism is associated with favorable prognosis in lung cancer, potentially due to its role in maintaining lung tissue homeostasis and preventing tumor progression [15,23]. These findings imply that patients like the Group^Low^, characterized by higher expression of surfactant metabolism-related genes, may benefit from targeted therapies aimed at preserving or enhancing these metabolic pathways. However, in this study, while these genes are associated with favorable prognosis, their specific functional roles in LUAD remain unclear.

Moreover, genomic profiling revealed that the Group^High^ exhibits frequent mutations in key genes such as *KEAP1*, *LYST*, *SETD2*, and *TP53*, which are involved in cell cycle regulation and DNA repair, along with copy number gains in oncogenes. These findings imply that CIN phenotype-related gene alterations could pave the way for targeted therapeutic interventions in LUAD, addressing the unmet need for more personalized treatment approaches based on molecular subtyping. Among them, *TP53* and *KEAP1* variants, in particular, have been linked to impaired cell cycle checkpoints and increased oxidative stress [2], both of which are hallmarks of CIN-driven tumorigenesis.

Integrative transcription factor (TF) profiling further highlighted the central role of FOXM1 and HMGA1 as key regulators of CIN phenotype-related genes in LUAD. Both transcription factors are implicated in cell cycle progression and chromatin remodeling [24], and their overexpression in high-grade LUAD tumors suggests that they may serve as potential therapeutic targets. Additionally, our findings demonstrated that the Group^High^, based on CNV analysis, exhibits gains of oncogenes, including *FOXM1*. It is plausible that the overexpression of *FOXM1* is associated with copy number gains, resulting in the upregulation of cell CIN phenotype-related genes and tumor progression.

Overall, our study highlights the clinical relevance of LUAD-significant CIN phenotype-related gene status, offering a potential framework for risk stratification and personalized treatment. The distinct transcriptomic and genomic landscapes of CIN phenotype-related subtypes provide a foundation for future studies aimed at validating these findings in clinical settings and exploring targeted therapies based on CIN phenotype status. However, further research is needed to confirm the functional roles of LUAD-significant CIN phenotype-related genes and their interactions in in vitro and in vivo studies.

## 5. Conclusions

In this study, we aimed to identify LUAD-significant CIN phenotype-related gene signatures associated with patient prognosis and established CIN phenotype-related genes as a key determinant of clinical outcomes in LUAD, highlighting their potential as prognostic markers. The identification of distinct CIN phenotype-related subtypes provides valuable insights into the molecular heterogeneity of LUAD and underscores the need for targeted therapeutic strategies to improve patient outcomes. Future studies should focus on validating these findings in larger cohorts and exploring the therapeutic potential of targeting CIN phenotype-related pathways in LUAD. 

## Figures and Tables

**Figure 1 cancers-16-03818-f001:**
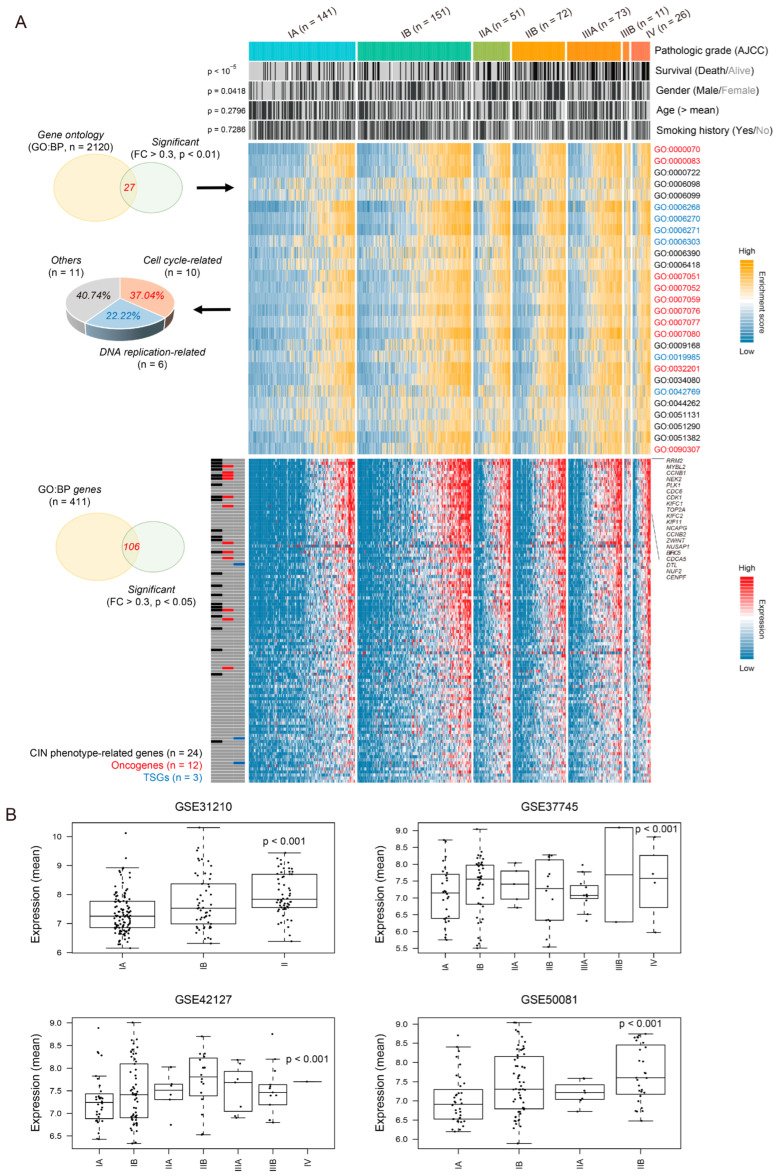
Chromosomal instability phenotype-related genes are associated with pathologic stage in lung adenocarcinoma. (**A**) Each sample is annotated with clinical information, including pathologic grade [American Joint Committee on Cancer (AJCC)], overall survival, gender, age, and smoking history (top). The heatmap shows the significant enrichment score of GO:BP (Gene Ontology: Biological Processes) terms (middle), with the associated genes for significant terms shown below (bottom). CIN-related genes, oncogenes, and tumor suppressor genes are indicated. The *p*-value was assessed using a *Chi*-squared test. (**B**) Boxplots show the mean values of LUAD-significant CIN-related genes across independent cohorts (*n* = 4). The *p*-value was assessed using one-way ANOVA.

**Figure 2 cancers-16-03818-f002:**
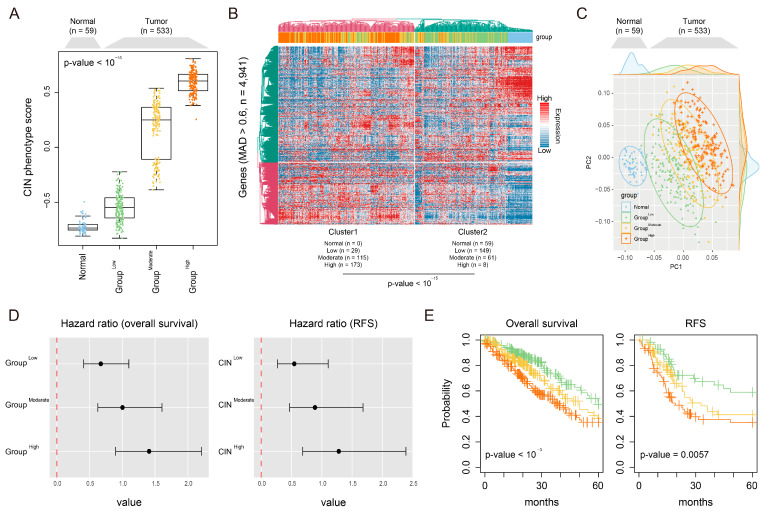
LUAD-significant CIN phenotype-related gene status revealed an association with transcriptomic patterns and prognostic predictions. (**A**) A Boxplot shows the enrichment score of LUAD-significant CIN-related genes (CIN score) across normal tissues and tumors. The *p*-value was assessed using one-way ANOVA. (**B**) A heatmap shows unsupervised clustering analysis using variably expressed genes (median absolute deviation, MAD > 0.6, *n* = 4941). The *p*-value was assessed using one-way ANOVA. (**C**) Principal component analysis using the variably expressed genes (median absolute deviation, MAD > 0.6, *n* = 4941) reveals that normal tissues and CIN groups are noticeably distributed. (**D**) Forest plots show the hazard ratio for OS and DFS across the CIN groups in the TCGA-LUAD dataset. (**E**) Kaplan–Meier curves of overall survival and disease-free survival for patients in the Group^Low^ (*n* = 176), Group^Moderate^ (*n* = 176), and Group^High^ (*n* = 181) from the TCGA-LUAD. The *p*-value was assessed using a log-rank test.

**Figure 3 cancers-16-03818-f003:**
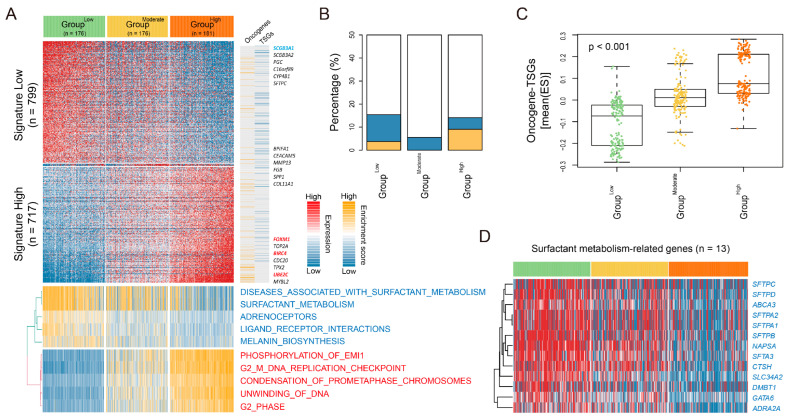
Characterization of transcriptomic subtypes according to CIN phenotype-related genes status. (**A**) A heatmap shows the expression patterns of differentially expressed genes (Signature Low, *n* = 799; Signature Moderate, *n* = 18; Signature High, *n* = 717). The top DEGs, oncogenes, and tumor suppressor genes (TSGs) are indicated on the right side (top), with the associated unsupervised gene signatures based on the reactome dataset shown below (bottom). (**B**) A barplot shows the proportion of the oncogenes and TSGs based on DEGs across CIN groups. (**C**) A Boxplot shows the oncoactivity scores [enrichment score (ES)_oncogenes_—ES_TSG_] across CIN groups. (**D**) A heatmap shows the expression pattern of surfactant metabolism-related genes (*p*-value < 0.01, one-way ANOVA, *n* = 13) across groups.

**Figure 4 cancers-16-03818-f004:**
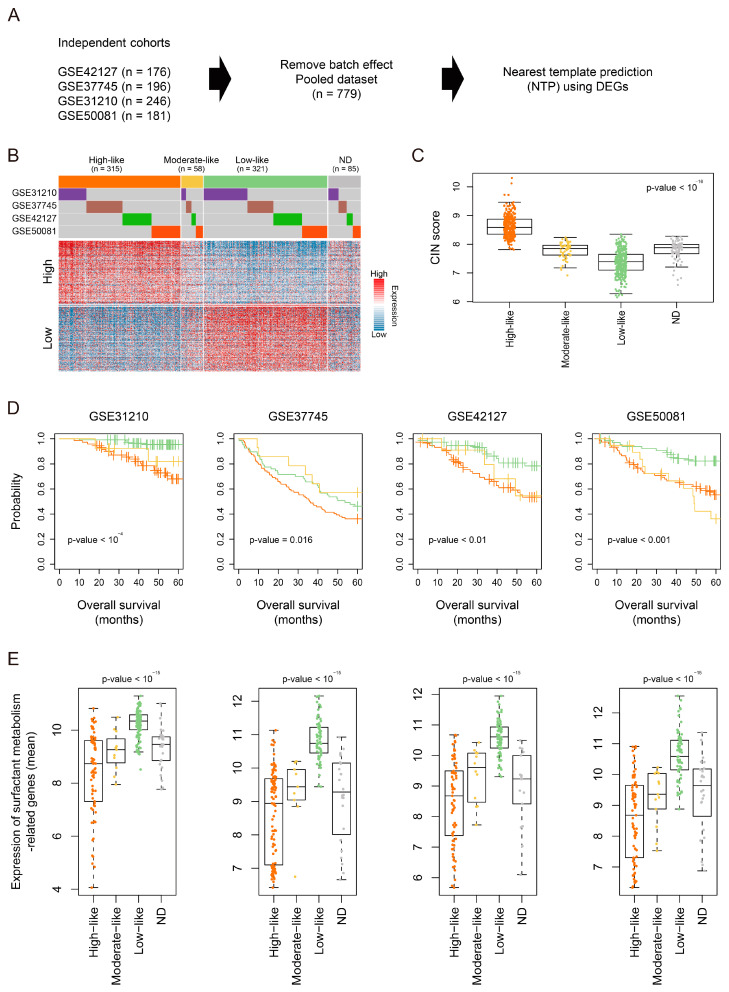
Validation of the molecular and clinical significance of LUAD-significant CIN phenotype-related gene status in a pooled LUAD dataset. (**A**) A schematic diagram illustrates the grouping [e.g., High-like, Moderate-like, Low-like, and ND (not determined) groups] of independent cohorts (*n* = 4) based on DEGs using nearest template prediction algorithm. (**B**) A heatmap shows the expression of DEGs across the groups in the pooled dataset (*n* = 779). (**C**) A Boxplot shows the oncoactivity scores across the groups in the pooled dataset. The *p*-value was assessed using one-way ANOVA. (**D**) Kaplan–Meier curves of overall survival for patients in the High-like, Moderate-like, and Low-like groups from each independent cohort. The *p*-value was assessed using a log-rank test. (**E**) Boxplots show the mean values of surfactant metabolism-related genes across independent cohorts (*n* = 4). The *p*-value was assessed using a one-way ANOVA.

**Figure 5 cancers-16-03818-f005:**
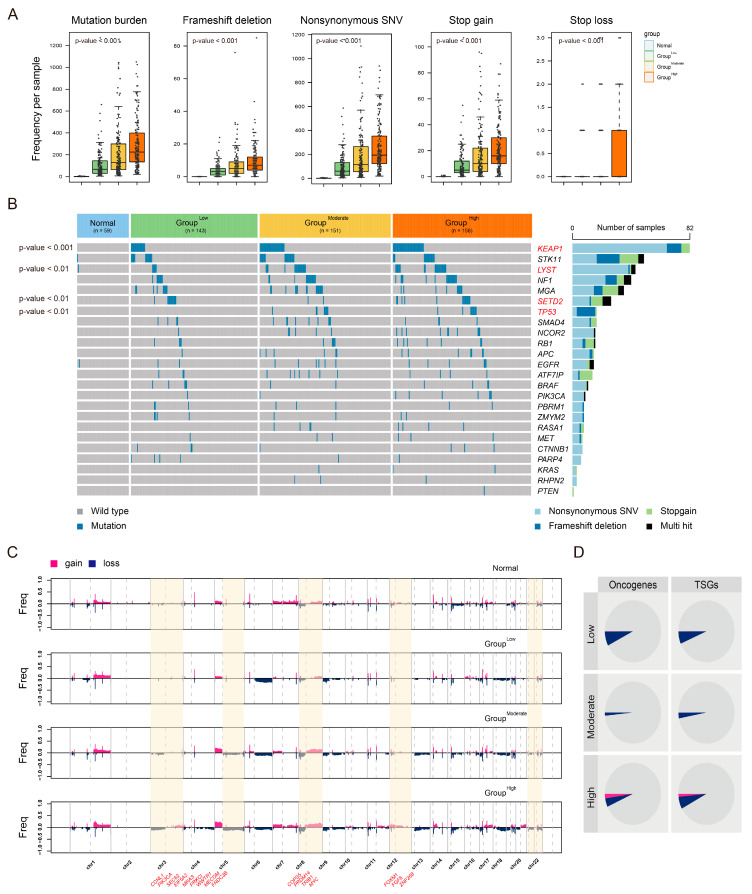
Distinct genomic profiles according to LUAD-significant CIN phenotype-related gene status. (**A**) Boxplots show the frequencies of mutation burden, frameshift deletion, nonsynonymous SNV, stop gain, and stop loss across normal tissue and CIN groups. The *p*-value was assessed using one-way ANOVA. (**B**) A heatmap shows the differentially mutated genes across normal tissue and CIN groups (left). Mutation frequencies are indicated (right). The *p*-value was assessed using a *Chi*-squared test. (**C**) Frequencies of copy number variation are shown in chromosomal order for each normal tissue and CIN group, with significant oncogenes indicated below. Significant gains and losses of CNV are highlighted in yellow. (**D**) Pie plots show the proportion of overlap between significant CNV genes and oncogenes (pink) or the TSGs (blue) in each group of TCGA-LUAD.

**Figure 6 cancers-16-03818-f006:**
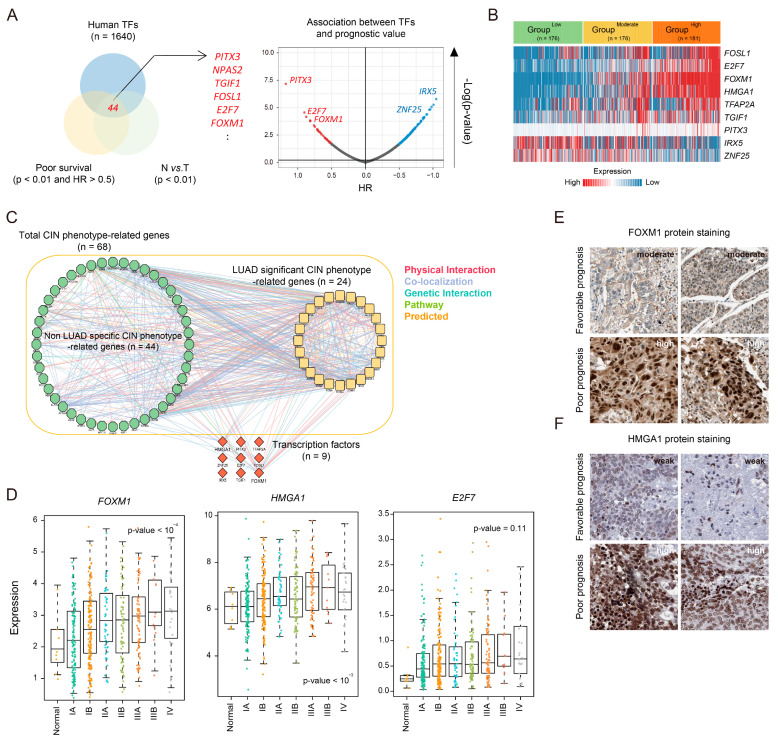
Transcription factor profiles according to LUAD-significant CIN phenotype-related gene status. (**A**) A Venn diagram shows the significant transcription factors (n = 44) associated with poor survival [*p*-value < 0.01 and hazard ratio (HR) > 0.5 compared to low expression of each transcription factor] and high expression of each transcription factor compared to normal tissue (*p*-value < 0.01 and fold change > 1) (left). A volcano plot shows transcription factors (n = 1640) based on prognostic value [−log(*p*-value) and HR]. Significant transcription factors are labeled in red (poor prognosis; *p*-value < 0.01 and HR > 0.5) and blue (good prognosis; *p*-value < 0.01 and HR < −0.5) (right). The *p*-value was assessed using a log-rank test. (**B**) A heatmap shows the significant transcription factors (n = 9, *p*-value < 0.001). The *p*-value was assessed using one-way ANOVA. (**C**) A genetic network of CIN-related genes (n = 68) was constructed using GeneMANIA software in Cytoscape (version 3.9.1), displaying physical interactions (pink), co-localization (purple), genetic interactions (sky blue), pathways (green), and predicted interactions (orange). LUAD-significant CIN phenotype-related genes (n = 24) and transcription factors (n = 9) are indicated. (**D**) Boxplots show the expression levels of the indicated transcription factors (n = 3, see Appendix A). The *p*-value was assessed using one-way ANOVA. (**E**,**F**) Images of immunohistochemical staining for FOXM1 and HMGA1 in LUAD patients (with well and poor prognosis), sourced from Protein atlas (https://www.proteinatlas.org/, accessed on 29 March 2024).

**Figure 7 cancers-16-03818-f007:**
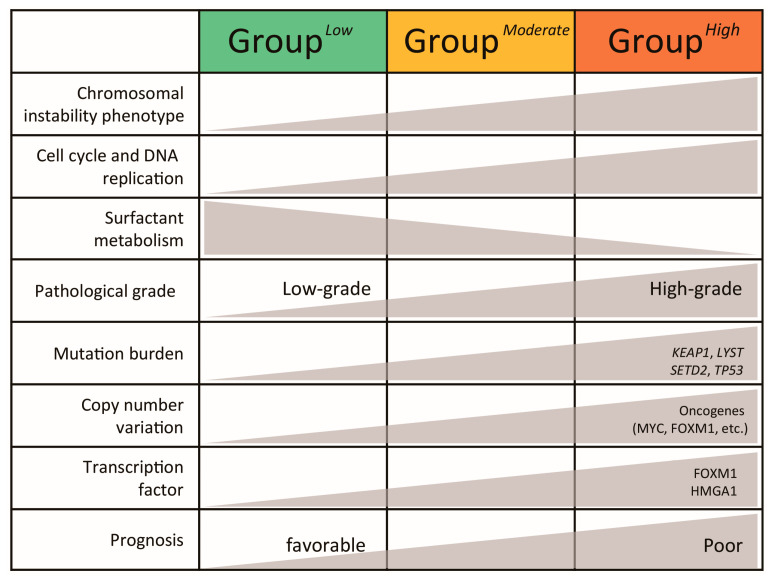
Summary of transcriptomic and genomic features of LUAD subtypes.

## Data Availability

The data generated and analyzed during the current study are available from the corresponding author upon reasonable request.

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
