# Peer review of "Identification of Molecular Subtypes and Prognostic Traits Based on Chromosomal Instability Phenotype-Related Genes in Lung Adenocarcinoma"

_cancers, 2024, doi:10.3390/cancers16223818_

Round 1

Reviewer 1 Report

Comments and Suggestions for Authors

In the manuscript “Identification of Molecular Subtypes and Prognostic Traits Based on Chromosomal Instability-Related Genes in Lung Adenocarcinoma,” Youngsic Jeon et al. performed a comprehensive transcriptomic analysis using TCGA-LUAD data, which included 59 cases of normal tissue and 533 cases of lung adenocarcinoma (LUAD). The authors identified 24 genes they termed “LUAD-specific CIN-related genes.” They evaluated the prognostic potential of these genes in predicting pathological grade and established their significance as key determinants of clinical outcomes in LUAD, highlighting their promise as prognostic markers. The findings were validated across multiple independent cohorts (779 cases).

The introduction of the manuscript provides a clear overview, including an accurate definition of chromosomal instability (CIN) as an increased rate of chromosomal gains and losses. However, concerns arise regarding the appropriateness of the term CIN throughout the manuscript. 

The manuscript lacks a clear definition of what constitutes “LUAD-specific CIN-related genes.”

The methodology for selecting the 24 “CIN” genes from the initial list of differentially expressed genes (DEGs) is not adequately explained.

A comprehensive list of DEGs is not provided, which could enhance the manuscript's transparency. Including this list in the Supplementary Materials would be beneficial.

The Gene Ontology (GO) Biological Process (BP) terms presented in Supplementary Table 1 indicate that many DEGs are associated with such processes as mitotic chromatin segregation, telomere maintenance, spindle organization, chromosome condensation, kinetochore assembly, etc. Given that approximately 400 genes in the human genome could be related to chromosomal instability, the criteria for the selection of only 24 CIN genes from the DEGs remains unclear.

Are these 24 genes definitely LUAD-specific or have they been identified in integrative bioinformatic studies of other cancer types as well?

Since the central theme of this study revolves around the significance of CIN-related genes as predictors of LUAD prognosis and their association with transcriptomic and genomic alterations, it is essential to provide a detailed explanation of how these genes were selected from the DEGs and what is their possible relation to CIN process (gains and losses of chromosomes during divisions of cancer cells).

Author Response

Comment 1

The introduction of the manuscript provides a clear overview, including an accurate definition of chromosomal instability (CIN) as an increased rate of chromosomal gains and losses. However, concerns arise regarding the appropriateness of the term CIN throughout the manuscript.

The manuscript lacks a clear definition of what constitutes “LUAD-specific CIN-related genes.”

The methodology for selecting the 24 “CIN” genes from the initial list of differentially expressed genes (DEGs) is not adequately explained.

A comprehensive list of DEGs is not provided, which could enhance the manuscript's transparency. Including this list in the Supplementary Materials would be beneficial.

The Gene Ontology (GO) Biological Process (BP) terms presented in Supplementary Table 1 indicate that many DEGs are associated with such processes as mitotic chromatin segregation, telomere maintenance, spindle organization, chromosome condensation, kinetochore assembly, etc. Given that approximately 400 genes in the human genome could be related to chromosomal instability, the criteria for the selection of only 24 CIN genes from the DEGs remains unclear.

 Are these 24 genes definitely LUAD-specific or have they been identified in integrative bioinformatic studies of other cancer types as well?

Since the central theme of this study revolves around the significance of CIN-related genes as predictors of LUAD prognosis and their association with transcriptomic and genomic alterations, it is essential to provide a detailed explanation of how these genes were selected from the DEGs and what is their possible relation to CIN process (gains and losses of chromosomes during divisions of cancer cells).

Response:

Upon re-evaluating our study and manuscript, we realized that the reference (Carter, S.L.; Eklund, A.C.; Kohane, I.S.; Harris, L.N.; Szallasi, Z. A signature of chromosomal instability inferred from gene expression profiles predicts clinical outcome in multiple human cancers. Nat. Genet. 2006, 38, 1043–1048.) for the database used to select the 24 genes was omitted, and we have now revised this section accordingly. Additionally, we would like to clarify that the 24 genes were not derived from DEGs. Instead, we identified 24 CIN-related genes that showed a significant association with increasing tumor grade, as presented in Figure 1A, and selected the top-ranking genes from this analysis to establish the set of 24 genes. However, based on your insightful recommendation, we have now included the full list of DEGs in the Supplementary Table 5 to enhance transparency.

Regarding these 24 genes definitely LUAD-specific CIN-related genes, we downloaded transcriptomic data from 31 cancer types available in the TCGA database and analyzed patient prognosis based on the expression levels of the 24 LUAD-specific CIN-related genes. Our results showed that these genes could predict patient outcomes in four cancer types (P < 0.001, ACC, LGG, MESO), including LUAD. Based on this, we consider these genes to be relatively significant to LUAD, so we have changed it to LUAD-significant CIN-related genes in manuscript. We greatly appreciate your thoughtful feedback. These results and statements have been added to the results and Figure S1 sections.

Reviewer 2 Report

Comments and Suggestions for Authors

The study identified chromosomal instability (CIN)-related genes that classify lung adenocarcinoma (LUAD) into three subtypes: CINLow, CINModerate, and CINHigh. The CINHigh group had significantly worse overall and disease-free survival, and CIN status was a stronger predictor of outcomes than pathologic grade. The CINHigh group also had frequent mutations in genes like KEAP1 and TP53. In contrast, the CINLow group had higher expression of surfactant metabolism-related genes, linked to better prognosis. CIN-related gene status emerged as a key prognostic marker, suggesting potential for therapeutic targeting in LUAD.

This is a well done study that can be published in Cancers.

At the same time, the extension of the analysis can further strengthen the study.  There are several small recommendations for the Authors:  : It will be good if they can фввкуыы at least some of them:

1. If it is possible, please test more gene signatures related to CIN. For example, it would be good to apply classical CIN70  (Carter, et al., 2006)

Carter, S.L.; Eklund, A.C.; Kohane, I.S.; Harris, L.N.; Szallasi, Z. A signature of chromosomal instability inferred from gene expression profiles predicts clinical outcome in multiple human cancers. Nat. Genet. 200638, 1043–1048. [Google Scholar] [CrossRef] [PubMed] 1329 citations

2. It also would be good to test a small TSG  -oncogene database consisting of 50 TSG and 50 oncogenes.

Davoli T.Xu A. W.Mengwasser K. E.Sack L. M.Yoon J. C.Park P. J., and Elledge S. J.Cumulative haploinsufficiency and triplosensitivity drive aneuploidy patterns and shape the cancer genomeCell. (2013155, no. 4, 948962https://doi.org/10.1016/j.cell.2013.10.011, 2-s2.0-84887992179.  850 citations

3. Perhaps the Authors can apply a classical data base TSG

Zhao M, Kim P, Mitra R, Zhao J, Zhao Z. TSGene 2.0: an updated literature-based knowledgebase for tumor suppressor genes. Nucleic Acids Res. 2016 Jan 4;44(D1):D1023-31. doi: 10.1093/nar/gkv1268. Epub 2015 Nov 20. PMID: 26590405; PMCID: PMC4702895.

Author Response

Comments 1: 

If it is possible, please test more gene signatures related to CIN. For example, it would be good to apply classical CIN70 (Carter, et al., 2006)

Carter, S.L.; Eklund, A.C.; Kohane, I.S.; Harris, L.N.; Szallasi, Z. A signature of chromosomal instability inferred from gene expression profiles predicts clinical outcome in multiple human cancers. Nat. Genet. 2006, 38, 1043–1048. [Google Scholar] [CrossRef] [PubMed] 1329 citations

Response 1:

In our study, we referenced chromosomal instability (CIN)-related genes based on the work by Carter et al. (2006), as you suggested. We acknowledge that this reference was not explicitly cited in the manuscript, and we have now revised it to include this important source. Thank you for your valuable suggestion.

Comments 2:

It also would be good to test a small TSG-oncogene database consisting of 50 TSG and 50 oncogenes.

Davoli T., Xu A. W., Mengwasser K. E., Sack L. M., Yoon J. C., Park P. J., and Elledge S. J., Cumulative haploinsufficiency and triplosensitivity drive aneuploidy patterns and shape the cancer genome, Cell. (2013) 155, no. 4, 948–962, https://doi.org/10.1016/j.cell.2013.10.011, 2-s2.0-84887992179.  850 citations

Response 2:

The TSG-oncogene database we used already includes most of the genes mentioned in the paper you cited (Davoli et al., 2013), so we were unable to conduct further analysis with an additional database. However, we appreciate your valuable suggestion, and we will consider incorporating this database in future analyses. Thank you for your insightful recommendation.

Comments 3: 

Perhaps the Authors can apply a classical data base TSG

Zhao M, Kim P, Mitra R, Zhao J, Zhao Z. TSGene 2.0: an updated literature-based knowledgebase for tumor suppressor genes. Nucleic Acids Res. 2016 Jan 4;44(D1):D1023-31. doi: 10.1093/nar/gkv1268. Epub 2015 Nov 20. PMID: 26590405; PMCID: PMC4702895.

Response 3:

Thank you for the suggestion to apply a classical TSG database. Our current analysis already integrates many of the well-established TSGs; however, we appreciate your recommendation and will consider incorporating additional classical databases in future studies. Your feedback is highly valued.

Round 2

Reviewer 1 Report

Comments and Suggestions for Authors

The authors have significantly improved the manuscript by addressing questions regarding the LUAD specificity of the 24-gene set used for identifying molecular subtypes and prognostic traits in lung adenocarcinoma. They downloaded transcriptomic data from 31 cancer types and analyzed patient prognosis based on the expression levels of these 24 genes. Their results indicate that these genes can also predict patient outcomes in three other cancer types, in addition to LUAD. Based on these findings, the authors now consider these genes to be relatively significant for LUAD and have revised the terminology to "LUAD-significant CIN-related" genes in the manuscript. These results and corresponding statements have been added to the Results section and Figure S1 (these changes must be done throughout the whole MS, for instance, on the Figure 6C and Supplementary Table 2 the set of 24 genes is still called LUAD-specific).

However, concerns regarding the appropriateness of the term “CIN (chromosomal instability)-related genes” remain unaddressed. The authors have included the previously omitted paper by Carter et al. (2006) and explained in their response that the 24-gene set was selected based on CIN70 genes identified by Carter et al. (2006).

In fact, the CIN70 score derived by Carter et al. cannot be used as a surrogate measurement of chromosomal instability but rather as a proxy for the specific patterns of the segmental aneuploidy assessed at the transcriptional level. In Carter et al. original publication, the term CIN70 refers to top 70 genes correlated with functional segmental aneuploidy, not chromosomal instability per se.  Assigning the term “CIN” to this set of genes has led to confusion among researchers, as “aneuploidy” and “chromosomal instability” are not interchangeable.

Furthermore, Carter et al. suggested in their paper: “The overexpression of the CIN signature in cells with high levels of aneuploidy may be due to at least three distinct mechanisms. It is possible that cells with aberrant DNA content produce more of the machinery required for chromosomal duplication and segregation. Alternatively, the expression level of the CIN signature may reflect a compensatory mechanism for impaired functioning of the machinery responsible for maintaining the integrity of genetic information at the chromosomal level. A third possibility is that the overexpression of the CIN genes allows cells to complete mitosis unimpeded by the usual checkpoints, thus conferring both direct proliferative advantage and increased cellular heterogeneity.” These proposed mechanisms are compensatory for chromosomal instability and are more related to the CIN phenotype (e.g., tolerance to CIN) than to chromosomal instability per se. 

In some publications, CIN has been used as an umbrella term, encompassing the presence of a CIN phenotype in cancer samples, including attributes like aneuploidy, tolerance, heterogeneity, instability, micronuclei, lagging chromosomes, multipolar mitoses, etc. Clarification of terminology with a clear indication of how the 24-gene set relates to the CIN process would greatly benefit this manuscript. If these 24 genes have no direct relation to chromosomal instability (i.e., the random chromosomal changes in tumor cells), and if CIN is used as an umbrella term, these assumptions must be clearly stated.

To enhance clarity and precision in the manuscript (MS), the authors should address these points to ensure accurate representation of their findings:

  1. Clarification on terminology: The term “CIN-related genes” needs a precise definition within the MS. The authors should clarify that these 24 genes are selected based on CIN70 signature proposed by Carter et al. (2006), and do not directly correspond to chromosomal instability (CIN) as a process. 
  2. Consider use of "CIN phenotype" as an umbrella term.
  3. Appropriate labeling of groups (CIN-low, CIN-medium, CIN-high): The authors should reconsider these terms, given that no direct CIN measurements were applied. If these classifications were based on the expression levels of genes, a clearer label (e.g., “gene expression-based stratifications”) would be more accurate.

Addressing these points would help align the MS with accurate scientific terminology, making the manuscript clearer and more precise for readers.

Author Response

Comments 1: Clarification on terminology: The term “CIN-related genes” needs a precise definition within the MS. The authors should clarify that these 24 genes are selected based on CIN70 signature proposed by Carter et al. (2006), and do not directly correspond to chromosomal instability (CIN) as a process. 

Answer 1: We sincerely thank you for your thorough and thoughtful guidance. We have clarified in the manuscript that the 24 genes referred to as "CIN-related" were selected based on the CIN70 signature proposed by Carter et al. (2006). We have added an explanation stating that these genes are associated with segmental aneuploidy at the transcriptional level rather than chromosomal instability (CIN) as process. This clarification has been included in the Results and Discussion sections.

Comments 2: Consider use of "CIN phenotype" as an umbrella term.

Answer 2: We appreciate this suggestion and have revised the manuscript to use "CIN phenotype" as an umbrella term that encompasses various cellular characteristics associated with CIN, including aneuploidy and tolerance to chromosomal aberrations. This terminology has been updated throughout the manuscript.

Comments 3: Appropriate labeling of groups (CIN-low, CIN-medium, CIN-high): The authors should reconsider these terms, given that no direct CIN measurements were applied. If these classifications were based on the expression levels of genes, a clearer label (e.g., “gene expression-based stratifications”) would be more accurate.

Answer 3: 

Thank you for this valuable feedback. We have revised the group labels to "low CIN phenotype expression," "moderate CIN phenotype expression," and "high CIN phenotype expression" to more accurately reflect that the classifications (GroupLow, GroupModerate, and GroupHigh) are based on stratifications of LUAD-significant CIN phenotype-related gene expression levels. These revisions have been made in the Results, Discussion, Figure legends, and Figure 2, 3, 5, 6, and 7.
